# Understanding Mental Health Status of Syrian Refugee and Jordanian Women: Novel Insights from a Comparative Study

**DOI:** 10.3390/ijerph20042976

**Published:** 2023-02-08

**Authors:** Fatin Atrooz, Sally Mohammad Aljararwah, Tzuan A. Chen, Omar F. Khabour, Samina Salim

**Affiliations:** 1Department of Pharmacological and Pharmaceutical Sciences, College of Pharmacy, University of Houston, Houston, TX 77204, USA; 2Department of Medical Laboratory Sciences, Faculty of Applied Medical Sciences, Jordan University of Science and Technology, Irbid 22110, Jordan; 3Department of Psychological Health and Learning Sciences, College of Education, University of Houston, Houston, TX 77204, USA; 4Health Research Institute, University of Houston, Houston, TX 77204, USA

**Keywords:** Syrian refugees, perceived stress, refugee education, refugee mental health, Jordan

## Abstract

(1) Background: War and displacement are well-known predictors of negative mental health outcomes among affected populations. This is especially relevant for refugees of war, particularly women, who often repress their mental health needs due to family responsibilities, social stigma, and/or cultural pressures. In this study, we compared the mental health status of urban Syrian refugee women (*n* = 139) with local Jordanian women (*n* = 160). (2) Methods: Psychometrically validated Afghan Symptom Checklist (ASC), Perceived Stress Scale (PSS), and Self-Report Questionnaire (SRQ) examined psychological distress, perceived stress, and mental health, respectively. (3) Results: According to independent *t*-tests, Syrian refugee women scored higher than Jordanian women on the ASC [mean score (SD): 60.79 (16.67) vs. 53.71 (17.80), *p* < 0.001], PSS [mean score (SD): 31.59 (8.45) vs. 26.94 (7.37), *p* < 0.001], and SRQ [mean score (SD): 11.82 (4.30) vs. 10.21 (4.72), *p* = 0.002]. Interestingly, both Syrian refugee and Jordanian women scored higher than the clinical cutoff in the SRQ. Regression analyses indicated that more educated women were less likely to score high on the SRQ (β = −0.143, *p* = 0.019), particularly in the anxiety and somatic symptoms subscale (β = −0.133, *p* = 0.021), and were less likely to exhibit symptoms of ruminative sadness (β = −0.138, *p* = 0.027). Employed women were more likely to exhibit high coping ability than unemployed women (β = 0.144, *p* = 0.012). (4) Conclusions: Syrian refugee women scored higher than Jordanian women in all used mental health scales. Access to mental health services and enhancing educational opportunities would help mitigate perceived stress and may enhance stress-coping abilities.

## 1. Introduction

The Syrian war, which began in 2011, is one of the world’s largest humanitarian crises in modern history and has resulted in the displacement of more than 6.7 million Syrians [1]. An estimated 1.4 million Syrians live in Jordan, of whom over 680,000 are registered refugees [2]. Since the average Syrian family size is 5.6, the estimated number of Syrian women refugees in Jordan is around a quarter of a million. In Jordan, Syrian refugees are allowed to work, with a majority of them working on a valid work permit and living on a government-issued residence permit [3,4]. In addition, most of the Syrian refugees in Jordan have access to education and health services, which are mainly offered by the UNHCR [5]. However, competing resources often fail to meet the complex health needs of this vulnerable group [2]. It is well-known that war-related displaced communities are often subjected to increased risks of persecution, discrimination, violence, resettlement, and adaptation problems in host communities [6,7]. This is not surprising considering that postmigration social stressors, including the complex asylum-seeking process, social support, and social isolation [8,9,10], and other factors such as poverty, limited employment opportunities, inadequate housing, and discrimination, are associated with mental health risks among refugees [11,12]. Such challenges can be overcome by implementing educational enrichment and other social support programs to help Syrian refugees develop resilience and recover from the negative outcomes of trauma, stress, and adversity [13]. In a review conducted to evaluate the primary health needs of displaced Syrians in Iraq, Jordan, Lebanon, Syria, and Turkey, mental health was identified as one of the greatest health concerns [14]. This is particularly relevant for Syrian refugee women, as they are considered to be highly vulnerable to adverse mental health outcomes [15,16,17]. In a recently published study, our data suggests that Houston-based Syrian refugee women in the United States (US) reported significantly higher mental stress and distress than Syrian refugee men, which potentially arises from the social strain of resettlement in a culturally novel environment in the US [17]. In addition, Syrian refugee girls were reported to be exposed to gender-based physical and psychological threats and abuses, along with the coercive practice of early marriage [18]. Therefore, it is important to note that context-specific conditions may impact mental health outcomes differently [19].

Studies examining the mental health of Syrian refugees in Jordan are limited. These include the psychological effects of the COVID-19 pandemic on refugee mental health, resilience, depression, and the history of trauma, barriers to the use of mental health services, and mental health symptoms [20,21,22,23]. In addition, some studies have investigated mental health among Jordanian women, which mainly focused on specific groups such as cancer patients, employed women, pregnant women, and victims of domestic violence and sexual abuse [24,25,26,27]. The present study was designed to examine the mental health and general well-being of Syrian refugee women who have resettled in the urban areas of the city of Ar-Ramtha in northern Jordan at the Syrian Jordanian border. Data collected from Syrian refugee women were compared with that of Jordanian women. As the association between mental health and physical well-being is well recognized [28,29], the presence of chronic illnesses was investigated. The impact of the new conditions in the host communities in Jordan, such as employment status and average income, were also examined as predictive variables of mental health outcomes. Other factors such as age, education, marital status, and the number of children were included in mental health outcome predictions, as these factors play a significant role in perceived stress and stress-coping abilities [19,30].

## 2. Materials and Methods

All communication forms and survey questionnaires utilized in the study were approved by the Institutional Review Board (IRB) Committee (STUDY00002929) for the Protection of Human Subjects, University of Houston (UH), Houston, TX, US, and by the Jordan University of Science and Technology (JUST) IRB, Irbid, Jordan (IRB#52/148/2022, 10 May 2022).

### 2.1. Subject Recruitment

Upon approval of the study protocol by the UH-IRB and JUST-IRB Committees, Syrian refugee and Jordanian women residing in the Ar-Ramtha area of northern Jordan were recruited to participate in the study using convenient and snowball sampling methods through the local Jordan community networks, including community organizations, charitable trusts, and social media groups. Snowball sampling was achieved by asking the recruited women to refer three potential subjects from their social network. Recruitment continued until the target number (about 300 participants) was achieved. Ar-Ramtha is a small city in the northern part of Jordan, which includes 7 small villages with a population of about 240 thousand people. The exact number of Syrian refugees who have settled in Ar-Ramtha is not known, but it is estimated that more than 40,000 Syrians currently reside in this area [31]. All Syrian refugees in the Ar-Ramtha area live within the local community. Both Syrian refugee women and Jordanian women were recruited from the same residential areas. The basic criteria for inclusion were adults, 18 years of age or older, Syrian refugee women, or Jordanian women. The postdoctoral fellow and the student researcher involved in the study are native Arabic speakers who explained the study objectives to the participants in their native language, Arabic, and obtained their consent. The target sample was calculated using G* Power 3.1.9.7. with α = 0.05, power = 0.95, and an effect size = 0.2; the required sample size was 262. A total of 450 women were invited to participate in the study, of whom 299 agreed to participate. The survey response rate was 66.5%.

### 2.2. Measures

Surveys included a sociodemographic questionnaire with general questions on the demographic and socioeconomic circumstances of the respondents, including age, education level, relationship status, number of children, number of family members residing in the same house, health insurance status, employment status, and average monthly income. Questions related to the prevalence of chronic diseases, including diabetes, hypertension, hypothyroidism, asthma, and irritable bowel syndrome, were included in the demographic characteristics section, as these measures were reported to be correlated with perceived stress and mental well-being [19,30,32,33]. Measures of perceived stress, mental health, and psychological distress included validated Arabic versions of the Perceived Stress Scale (PSS), Self-Reporting Questionnaire (SRQ), and Afghan Symptoms Checklist (ASC), respectively. The PSS questionnaire is the most widely used psychological instrument for measuring the degree to which situations in one’s life are appraised as stressful [17,34]. The PSS instrument consists of 7 positive items, which represent the coping ability subscale, and 7 negative items, which represent perceived distress. The Arabic version of the PSS-14 has been previously validated [17,35]. The PSS-14 showed adequate reliability in our sample, with a Cronbach’s alpha coefficient of 0.76 for the full scale and 0.77 and 0.81 for the negative subscale and coping ability subscale, respectively.

The SRQ is an instrument developed by the World Health Organization to screen for mental disorders, including depression, anxiety-related disorders, and somatoform symptoms [36]. It consists of a questionnaire comprising 24 questions: 20 questions related to neurotic symptoms and 4 questions related to psychotic symptoms [37]. A score of 7 or above indicates the presence of a potential psychological problem. In this study, we used the short form of the SRQ (SRQ-20), which consists of the first 20 non-psychotic items, as this instrument was previously validated in the Arab population [38,39]. Items in the SRQ were classified into two factors (subscales) following the Idaiani et al. study [40]. Factor 1 was identified as depression, while factor 2 was a symptom of anxiety. The depression factors consist of trouble thinking clearly, unhappiness, crying more, difficulty enjoying an activity, difficulty making decisions, daily work suffering, inability to play a useful part in life, losing interest in things, easily becoming tired, and a feeling of worthlessness. The somatic-anxiety factor consists of poor appetite, poor sleep, being easily frightened, hands shaking, being tense and worried, poor digestion, headaches, and being tired all the time. In our sample, the SRQ-20 also showed high reliability, with a Cronbach’s alpha value of 0.84.

The ASC was developed and used in Kabul, Afghanistan, to identify local indicators of psychological distress in conflict and post-conflict situations [41]. The Arabic version of the ASC instrument demonstrated excellent reliability in our previous study conducted with Arab American refugees and immigrants who resettled in Houston, TX, USA [17]. The ASC is a 22-item instrument that asks about one’s feelings and experiences over a 2-week period. The ASC questionnaire consists of three interpretable subscales: (1) sadness with social withdrawal and somatic distress, (2) ruminative sadness without social isolation and somatic distress, and (3) aggression or stress-induced reactivity, which is indicated by quarreling, beating one’s children, and nervousness [42]. The instrument demonstrated excellent reliability with a Cronbach’s alpha value of 0.90.

### 2.3. Online Survey

The Arabic versions of the survey questionnaires were uploaded on the REDCap platform, which enables the secure building and management of online surveys. The study team used the survey link to access the questions while interviewing the participants. The research team obtained the participants’ consent electronically via a REDCap survey link. Participants had the option to complete the survey questionnaire either independently or with one-on-one guidance from the postdoctoral fellow or the student researcher. The study team explained the questions if the participants asked for clarification. Upon survey completion, each participant received a $14 (10 Jordanian Dinars) gift card.

### 2.4. Statistical Analysis

The statistical analyses were conducted using complete case analyses, and the response rates of variables of interest ranged from 96% to 100%. Data were first examined using descriptive statistics. Sample comparisons between Syrian and Jordanian women were performed using an independent *t*-test or chi-square test for continuous and categorical variables, respectively. An analysis of covariance (ANCOVA) was used to compare the scores on each stress scale between Syrian and Jordanian women while controlling for participants’ age, employment status, and relationship status. An ANCOVA was conducted separately for each stress and mental health scale. The effects of participant characteristics on each stress scale were examined using stepwise linear regression to identify predictor variables. Separate regression analyses were conducted for each stress scale. The correlations between perceived stress and mental health measures were also examined using person-correlation analyses. Significance was set at *p* < 0.05. All analyses were conducted using IBM SPSS statistics (version 29.0).

## 3. Results

Data were collected from a total of 139 Syrian refugee women and 160 Jordanian women residing in the Ar-Ramtha area of northern Jordan between July and September 2022.

### 3.1. Demographic Characterestics

Table 1 depicts the demographic characteristics and socioeconomic circumstances of the participants. Participants’ ages ranged from 18 to 77 years (mean = 41.36). The age distribution was comparable among Syrian and Jordanian women (*p* = 0.207). Education level was significantly different between Syrian refugee and Jordanian women (*p* < 0.001), with 84.9% of Syrian refugee women not completing high school compared to 48.8% of Jordanian women. While 81.9% of Jordanian women reported having health insurance, only 6.5% of refugee women had health insurance (*p* < 0.001). Additionally, 97.8% of Syrian refugees had a monthly income below 551 JD (~750 $), which is significantly higher than 81.9% of Jordanians who reported a monthly income below 551 JD (*p* < 0.001). The percentage of Syrian refugee women who reported living with more than one family in the same house (38.8%) was significantly higher (*p* < 0.001) than Jordanian women (10.6%).

### 3.2. Perceived Stress and Mental Health Measures

In the PSS, Syrian refugee women scored significantly higher than Jordanian women (*p* < 0.001), as seen in Figure 1A. Further analysis of positive and negative items within the PSS revealed that Syrian refugee women reported higher scores in the perceived distress subscale, negative items (*p* = 0.004), and lower scores in the coping ability subscale, positive items (*p* < 0.001), compared to Jordanian women, as seen in Table 1. The results of the ANCOVA revealed a similar significant difference between Syrian refugee and Jordanian women in the PSS total score when controlling for participants’ age, employment status, and relationship status (*p* < 0.001).

In the SRQ, Syrian refugee women reported a higher total score compared to Jordanian women (*p* = 0.002), see Figure 1B. Although both reported above the clinical cutoff score of 7, subscales analysis within the SRQ revealed that Syrian refugees exhibited significantly higher depressive symptoms compared to Jordanian women (*p* < 0.001). While anxiety and somatic symptoms were comparable between the two groups (*p* = 0.135), as seen in Table 1, the results of the ANCOVA revealed a significant difference between Syrian refugee and Jordanian women in the total SRQ score when controlling for participants’ age, employment status, and relationship status (*p* = 0.010).

In ASC, Syrian refugees reported a significantly higher total score compared to Jordanian women (*p* < 0.001), see Figure 1C. Subscales analysis within the ASC measure revealed that Syrian refugees reported higher scores than Jordanian women in the sadness with social withdrawal subscale (*p* < 0.001) as well as in the ruminative sadness subscale (*p* = 0.002), but not in the aggression subscale (*p* = 0.286), as seen in Table 1. An ANCOVA analysis revealed a significant difference between Syrian refugee and Jordanian women in the total ASC score when controlling for participants’ age, employment status, and relationship status (*p* < 0.001).

### 3.3. Linear Regression Analyses

Linear regression analysis indicated that more educated women were less likely to score highly in the SRQ (*p* = 0,019), particularly in the anxiety and somatic symptoms subscales (*p* = 0.021), and they were also less likely to exhibit ruminative sadness (*p* = 0.027). Employed women were more likely to exhibit high coping ability as compared to unemployed women (*p* = 0.012). Women with health insurance were less likely to report high perceived distress than women without health insurance (*p* = 0.001). Unmarried, divorced, or widowed women were more likely to report higher scores in the SRQ than married women (*p* = 0.022). Women with hypertension were more likely to report higher scores than women without hypertension in the PSS. Women with hypertension were also more likely to report high scores in the SRQ (*p* = 0.018) and the ASC (*p* = 0.002). Women with irritable bowel syndrome were more likely to report high scores in the SRQ (*p* = 0.018) than women without irritable bowel syndrome. Women with more children were more likely to report a high aggression score (*p* = 0.018) than women with fewer children, as seen in Table 2.

### 3.4. Correlation Analyses

Further correlation analysis between mental health measures and PSS total score and subscales showed that perceived stress and perceived distress were positively correlated with SRQ total scores and depression and anxiety subscales, as well as with ASC total score and ASC subscales, as seen in Table 3. Coping ability was negatively correlated with the SRQ total score and the depression and anxiety subscales. Coping ability was also negatively correlated with the ASC total score, particularly with sadness on the social withdrawal subscale, as seen in Table 3.

## 4. Discussion

Forced displacement and migration are well-recognized stressors that often contribute to complex mental health issues among refugees [6,43,44], with a reportedly high vulnerability to mental distress and greater susceptibility to mental illnesses among women refugees [44,45,46,47]. This is not surprising considering the risks, violence, and exploitation that women often face during displacement and resettlement processes [48,49]. While the impact of displacement stressors on women refugees is well documented [50,51,52,53], context-specific factors that contribute to mental distress are not well understood. This study provides important information on context-specific issues of mental stress among a sample of Syrian refugee women resettled in an urban setting in northern Jordan. The levels of perceived stress among Syrian refugee women in our sample are predictive of important mental health outcomes, as discussed below. While pre-migration traumatic stress, such as torture, loss of property, physical assault, and loss of livelihood, is recognized as a key predictor of mental health outcomes in refugees and asylum seekers [54], recent research has focused on the psychological impact of post-migration stressors in the novel environment of resettlement [11]. The post-resettlement experience of Syrian refugees resettled in Jordan is typically characterized by extreme poverty and uncertainty around basic needs such as housing, health care access, education, and employment [55]. The post-migration environment is especially critical for refugee women as they are considered a highly vulnerable group, often with worse trauma-related mental health outcomes [51,56].

In our sample, Syrian refugee women reported high levels of perceived stress compared to the local Jordanian women. Correlation analysis revealed that hypertension was a predictable variable of increased perceived stress. This is concerning, as having no health insurance was also a predictable variable of perceived distress. Considering that approximately 93.5 % of Syrian refugee women were without medical insurance, a lack of health insurance or access is a predictable risk factor for perceived distress and associated mental health outcomes in Jordan-based Syrian refugee women. This is not surprising considering Syrian refugees’ lack of access to adequate healthcare in Jordan. The healthcare challenges of Syrian refugees in Jordan are not just about disparities in access and outcomes; they have become an economic, political, and social crisis for the Jordanian government [57]. Furthermore, the coping ability was significantly higher among Jordanian women when compared to Syrian refugee women. Linear regression analysis suggested that coping ability was negatively correlated with having diabetes but positively correlated with having a job. Although the percentages of both predictable variables, employment and having diabetes, are comparable between Syrian refugee and Jordanian women, other factors such as financial hardship may negatively impact coping abilities among Syrian refugee women. It has been reported that low socioeconomic status and income are important determinants of the refugee coping process [58]. Another significant factor affecting coping ability is education. A literature review that focused on East African refugees indicated that education is an important factor that improves refugees’ coping abilities and that, through education, young refugees sought hope for a safe, prosperous, and equitable future [59]. The school enrollment rate for Syrian refugees in Jordan is less than 60% [60]. Low enrollment rates also suggest that refugees face challenges in continuing and completing basic education in Jordan [61]. Only a small fraction of refugees (8%) are currently enrolled in higher education in Jordan. As the opportunities for higher education are limited for Syrian refugees and considering the access to these opportunities is unequal, this often results in refugees not being able to see the value of education [62]. This limited opportunity for education is particularly concerning for girls, as early marriage and school dropout are interrelated outcomes that have an enormous impact on refugee girls’ vulnerability to adverse mental health outcomes [18]. Mental health stigma remains one of the most acknowledged reasons why refugees fail to access mental health services. Many studies have previously reported the underutilization of mental health services by various refugee groups when compared to the general population [63,64,65]. For example, Afghan refugee women [66,67] in the United States and Syrian refugee women in Lebanon [68], Canada [69], and Germany [70] are reported to have unmet mental health needs and low use of mental health services. Lack of knowledge about mental health is one of the major barriers to seeking mental health, highlighting the significance of refugee education as an empowering tool [71].

We observed that Syrian refugee women exhibited significantly higher scores in the SRQ than Jordanian women, particularly in the depression subscale. Syrian refugee women also exhibited higher scores in ASC than Jordanian women, particularly in the sadness with social withdrawal and ruminative sadness subscales. Pearson’s correlation analyses revealed that perceived stress and distress have a significant positive correlation with all mental health outcomes. While pre-migration traumatic stress is recognized as a key predictor of mental health outcomes in refugees, post-migration stressors in the settlement environment are critical determinants of refugee mental health [11]. The stressful post-migration environment is especially concerning for refugee women as a vulnerable group to mental health issues [14,72]. As wives and/or mothers, women bear extra burdens in the process of immigration in order to support family members as they adjust to a new way of life and often undertake the role of protecting and upholding family values, culture, and beliefs [73,74]. It is significant to point out here that both Jordanian and Syrian refugee women scored higher than the clinical cutoff of 7 on the SRQ scale. Women are the primary caregivers in the family and often ignore their own healthcare needs to protect the family, which may partially explain their vulnerability to psychological and mental health symptoms [75]. In fact, mental health is still highly stigmatized in Arab culture [76]. Additionally, only a few mental health institutions exist in Jordan; there are only three mental health hospitals for adults and one specialized psychiatric hospital for children [77,78], with the total number of practicing psychiatrists in Jordan not exceeding 2 per 100,000 residents [79]. Thus, the stigma around mental health combined with the scarcity of mental health care institutions is suggested as the main barrier to seeking mental health care among both Syrian refugees as well as Jordanian women [79].

In our study, the number of children was a predictive variable for women reporting an aggression score. While the prediction of aggression was comparable between Syrian refugees and Jordanian women, the lack of social support, the worrying about family members left behind in Syria, poverty, and the limited resources might exacerbate the coping strategies of Syrian refugee women.

## 5. Conclusions

To effectively serve resettlement and facilitate the effective integration of refugees into host communities, mental health professionals and policymakers must be cognizant of the impact of the refugee experience and cultural contexts on psychosocial functioning. Social workers and health professionals should consider designing and implementing targeted programs that address current stressors and rebuild indigenous social support systems to enhance acculturation and reduce mental health problems among Syrian refugee women.

## 6. Limitations

Our study has several limitations. First, the reliance on self-report, as opposed to clinical interview; second, the study focused on one city. Therefore, the results cannot be generalized to other similar populations resettling in other cities in Jordan. Third, the cross-sectional nature of our observations mitigates cause-and-effect conclusions. Finally, the questionnaires did not include all the factors that may impact women’s mental health.

## Figures and Tables

**Figure 1 ijerph-20-02976-f001:**
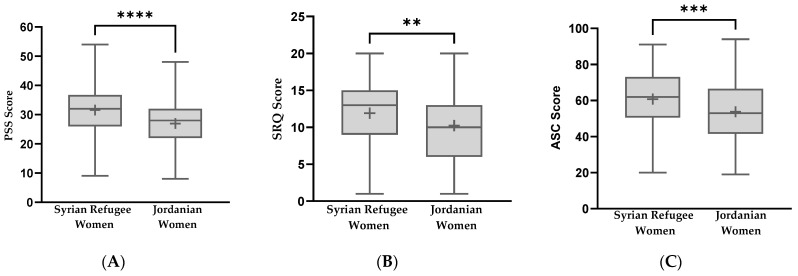
Comparison between Syrian refugee and Jordanian women in perceived stress and mental health measures. (**A**) Perceived Stress Scale (PSS) total score, (**B**) Self-Reporting Questionnaire (SRQ), and (**C**) Afghan Symptoms Checklist (ASC). Means are represented by (+) symbol, whiskers represent minimum and maximum values. Data were analyzed using an independent *t*-test. ** Significantly different at *p* < 0.01, *** Significantly different at *p* < 0.001, **** Significantly different at *p* < 0.0001.

**Table 1 ijerph-20-02976-t001:** Sociodemographic characteristics and mental health scale scores of Syrian refugee and Jordanian women.

Variable	Syrian Refugee Women (*n* = 139) N [%]/Mean (SD)	Jordanian Women (*n* = 160) N ([%]/Mean (SD))	Total (*n* = 299) N [%]/Mean (SD)	Statistics	*p*-Value
Age					
	40 (13.3)	42.23 (13.4)	41.36 (13.4)	1.202	0.115
Age Group					
18–25	27 [20.5]	19 [12.4]	46 [16.1]	4.565	0.207
26–39	34 [25.8]	44 [28.8]	78 [27.4]		
40–59	64 [48.5]	73 [47.7]	137 [48.1]		
60–77	7 [5.3]	17 [11.1]	24 [8.4]		
Education					
Less than High School	118 [84.9]	78 [48.8]	196 [65.6]	43.451	<0.001 ***
High School	13 [9.4]	44 [27.5]	57 [19.1]		
College/University Degree	8 [5.8]	38 [23.7]	46 [15.4]		
Relationship Status					
Single	9 [6.5]	17 [10.6]	26 [8.7]	3.611	0.307
Married	104 [74.8]	122 [76.3]	226 [75.6]		
Divorced	9 [6.5]	5 [3.1]	14 [4.7]		
Widowed	17 [12.2]	16 [10.0]	33 [11.0]		
Average Monthly Income					
Less than 250 JD	93 [67.4]	55 [34.4]	148 [49.7]	40.2666	<0.001 ***
250–550 JD	42 [30.4]	76 [47.5]	118 [39.6]		
551–800 JD	3 [2.2]	14 [8.8]	17 [5.7]		
800–1500 JD	0 [0.0]	11 [6.9]	11 [3.7]		
More than 1500 JD	0 [0.0]	4 [2.5]	4 [1.3]		
Number of Children					
0–2	35 [25.5]	92 [59.0]	127 [43.3]	33.293	<0.001 ***
3–7	96 [70.1]	61 [39.1]	157 [53.6]		
8+	6 [4.4]	3 [1.9]	9 [3.1]		
Employment					
Yes	26 [18.7]	32 [20.3]	58 [19.5]	0.113	0.737
No	113 [81.3]	126 [79.7]	239 [80.5]		
Live With More Than One Family in The Same House					
Yes	54 [38.8]	17 [10.6]	71 [23.7]	32.72	<0.001 ***
No	85 [61.2]	143 [89.4]	228 [76.3]		
Health Insurance					
Yes	9 [6.5]	131 [81.9]	140 [46.8]	169.834	<0.001 ***
No	130 [93.5]	29 [18.1]	159 [53.2]		
Diabetes					
Yes	20 [14.4]	23 [14.4]	43 [14.4]	0.000	0.997
No	119 [85.6]	137 [85.6]	256 [85.6]		
Hypertension					
Yes	30 [21.6]	32 [20.0]	62 [20.7]	0.113	0.736
No	109 [78.4]	128 [80.0]	237 [79.3]		
Hyperthyroidism					
Yes	8 [5.8]	11 [6.9]	19 [6.4]	0.157	0.692
No	131 [94.2]	149 [93.1]	280 [93.6]		
Asthma					
Yes	8 [5.8]	5 [3.1]	13 [4.3]	1.237	0.266
No	131 [94.2]	155 [96.9]	286 [95.7]		
Irritable Bowel Syndrome					
Yes	31 [22.3]	35 [21.9]	66 [22.1]	0.008	0.929
No	108 [77.7]	125 [78.1]	233 [77.9]		
Perceived Stress Scale (PSS)					
Total PSS score (range: 0–56)	31.59 (8.45)	26.94 (7.37)	29.08 (8.21)	5.035	<0.001 ***
Subscale: Perceived Distress (0–28)	18.51 (5.45)	16.64 (5.64)	17.50 (5.6)	2.88	0.004 **
Subscale: Coping Ability (0–28)	14.13 (5.60)	16.81 (5.72)	15.58 (5.81)	4.05	<0.001 ***
Self-Reporting Questionnaire (SRQ)					
Total SRQ Score (range: 0–20)	11.82 (4.30)	10.21 (4.72)	10.96 (4.59)	3.054	0.002 **
Subscale: Depression (range: 0–10)	6.02 (2.77)	4.82 (2.72)	5.38 (2.81)	3.753	<0.001 ***
Subscale: Anxiety and Somatic Symptoms (range: 0–10)	5.80 (2.21)	5.39 (2.46)	5.58 (2.35)	1.511	0.135
Afghan Symptoms Checklist (ASC)					
Total ASC score (range: 20–100)	60.79 (16.67)	53.71 (17.80)	56.96 (17.62)	3.473	<0.001 ***
Sadness with Social Withdrawal (range: 15–75)	46.56 (13.37)	40.71 (13.99)	43.40 (13.99)	3.616	<0.001 ***
Fishar-Ruminative Sadness (range: 2–10)	6.85 (2.42)	5.87 (2.72)	6.32 (2.63)	3.204	0.002 **
Aggression (range: 4–20)	10.64 (3.68)	10.17 (3.72)	10.39 (3.70)	1.071	0.286

Note. Data were analyzed using chi-square for categorical variables and t-test for continuous variables. JD = Jordanian Dinar; ** *p* < 0.01; *** *p* < 0.001.

**Table 2 ijerph-20-02976-t002:** Linear regression analyses for perceived stress and mental health measures and participants’ sociodemographic characteristics.

Variable	Significant Predictors	β	SE	*p*-Value
Perceived Stress Scale	Syrian Refugee Women (ref Jordanian)	0.274	0.927	<0.001
	Hypertension (ref No)	0.116	1.142	0.004
Perceived Distress	Health Insurance (ref No)	−0.190	0.658	0.001
	Hypertension (ref No)	0.124	0.811	0.033
Coping Ability	Syrian Refugee Women (ref Jordanian)	−0.227	0.661	<0.001
	Employment Status (ref No)	0.144	0.832	0.012
	Diabetes (ref No)	−0.129	0.942	0.025
Self-Reporting Questionnaire	Syrian Refugee Women (ref Jordanian)	0.127	0.547	0.034
	Relationship Status (ref Married)	0.130	0.602	0.022
	Education Level (Scale)	−0.143	0.167	0.019
	Irritable Bowel Syndrome (ref No)	0.626	0.134	0.018
	Hypertension (ref No)	0.648	0.136	0.018
Depression	Syrian Refugee Women (ref Jordanian)	0.189	0.340	0.002
	Education Level	−0.115	0.104	0.066
	Age (Scale)	0.186	0.012	0.002
Anxiety and Somatic Symptoms	Relationship Status (ref Married)	0.139	0.314	0.016
	Education Level (Scale)	−0.133	0.081	0.021
	Irritable Bowel Syndrome (ref No)	0.193	0.327	<0.001
Afghan Symptoms Checklist	Syrian Refugee Women (ref Jordanian)	0.197	2.050	<0.001
	Hypertension (ref No)	0.181	2.523	0.002
Sadness with Social Withdrawal	Syrian Refugee Women (ref Jordanian)	0.202	1.629	<0.001
	Hypertension (ref No)	0.165	2.005	0.005
Ruminative Sadness	Syrian Refugee Women (ref Jordanian)	0.132	0.326	0.033
	Education Level (Scale)	−0.138	0.097	0.027
Aggression	Hypertension (ref No)	0.208	0.543	<0.001
	Number of Children	0.168	0.402	0.005

**Table 3 ijerph-20-02976-t003:** Correlation analyses for mental health scores and perceived stress and coping ability.

Mental Health Measure		Perceived Stress Scale	Subscale- Perceived Distress	Subscale- Coping Ability
Self-Reporting Questionnaire	r	0.557 ***	0.525 ***	−0.272 ***
	*p*	<0.001	<0.001	<0.001
Depression	r	0.575 ***	0.511 ***	−0.317 ***
	*p*	<0.001	<0.001	<0.001
Anxiety and Somatic Symptoms	r	0.402 ***	0.414 ***	−0.152 **
	*p*	<0.001	<0.001	0.009
Afghan Symptoms Checklist	r	0.583 ***	0.689 ***	−0.119 *
	*p*	<0.001	<0.001	0.043
Sadness with Social Withdrawal	r	0.580 ***	0.685 ***	−0.116 *
	*p*	<0.001	<0.001	0.049
Fishar-Ruminative Sadness	r	0.362 ***	0.445 ***	−0.042
	*p*	<0.001	<0.001	0.478
Aggression	r	0.470 ***	0.566 ***	−0.089
	*p*	<0.001	<0.001	0.128

Note. * *p* < 0.05; ** *p* < 0.01; *** *p* < 0.001.

## Data Availability

The data presented in this study are available upon request from the corresponding author. The data are not publicly available due to privacy.

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
