# Peer review of "Understanding Mental Health Status of Syrian Refugee and Jordanian Women: Novel Insights from a Comparative Study"

_ijerph, 2023, doi:10.3390/ijerph20042976_

Round 1

Reviewer 1 Report

Important topic that deserves some attention

Appreciate the greater context of women’s issues as a backdrop to refugee-specific challenges

The paper is succinctly written, which is appreciated.

There are a handful of grammatical, sentence structure and paragraph structure issues that may be best addressed by having a professional writer of English review the paper.

The introduction is good in content and quantity of writing

Methods

The choice of the Afghan Symptom Checklist seems curious as there are idioms of distress that are particular to the context of Afghan culture and languages. You highlight validation in Arabic based on your previous work. It would be helpful to understand how this tool is valid for a different culture, particularly given the specific Afghan idioms. Can you clarify this for the reader?

The statistical analysis seems appropriate and well-reported through the paper

Results

You later comment on the small numbers but, in my view, this is a good number of participants for a study with refugees.  

Demographic measures seem relevant to the context.

You report “significant” differences, which appears statistically true, but the differences are not that large. Can you discuss this a bit more? This study may say almost as much about non-refugee women as it does about refugees.

Tables 2 and 3. Can this be depicted in graphic form? It will be difficult for most readers to sift through the statistics and make sense of it. A visual snapshot of what these numbers mean would be helpful.

Discussion

Overall good and focussed enough to capture the essential points.

Good comments on the importance of health care insurance.

The sentence: “When opportunities of higher education…value of education” is bit confusing. I suggest reworking this sentence structure.

I am glad you highlight that both the refugee and host women score high on the SRQ

You highlight a number of women’s issues but do not comment on sex and gender based violence. Although not the focus of your study, you may wish to acknowledge it in your introduction and/or discussion.

One of the limitation of your study is that it cannot capture all the relevant factors affecting women’s mental health.

References seem to draw on culturally relevant literature  

Author Response

RESPONSE LETTER

We are thankful to the reviewers for their thoughtful and very detailed review of our manuscript. All the recommended changes and edits have been incorporated in the revised version of our manuscript.  We greatly appreciate the time and effort invested to produce this detailed review. Our point-by-point response is provided below.

REVIEWER#1

Important topic that deserves some attention. Appreciate the greater context of women’s issues as a backdrop to refugee-specific challenges. The paper is succinctly written, which is appreciated. There are a handful of grammatical, sentence structure and paragraph structure issues that may be best addressed by having a professional writer of English review the paper. The introduction is good in content and quantity of writing.

Response: Thank you. As suggested, the manuscript has been read and edited by a native English speaking colleague of the senior author of this manuscript at the University of Houston.

Methods

  1. The choice of the Afghan Symptom Checklist seems curious as there are idioms of distress that are particular to the context of Afghan culture and languages. You highlight validation in Arabic based on your previous work. It would be helpful to understand how this tool is valid for a different culture, particularly given the specific Afghan idioms. Can you clarify this for the reader?

Response: Although Arab and Afghan cultures are different, they have several commonalities such as religious practices and patriarchal family structure with men in the family, as the breadwinner of the household, while the role of the women is that of a homemaker who takes overall responsibility of the household work. As reported in our previous paper (PMID: 35270240), we replaced Afghan specific idioms with Arabic idioms that reflect the same meaning. The following paragraph was included in our previous published paper which we referred to in the present paper:

“Among the items are three Dari terms (Dari is the native language spoken by people from Afghanistan) representing Afghan idioms of distress: jigar khun, a term describing a form of sadness that includes grief following interpersonal loss; asabi, a term for feeling nervous or highly stressed, and fishar, or “blood pressure”, which refers to internal agitation or low energy and motivation. These terms were translated into Arabic utilizing culturally appropriate terms”

  1. The statistical analysis seems appropriate and well-reported through the paper Response: Thank you.

Results

  1. You later comment on the small numbers but, in my view, this is a good number of participants for a study with refugees. Demographic measures seem relevant to the context.

  Response: Thank you. We added sample size calculation and power analysis in the methods section as suggested by other reviewers. The following sentences were added to the revised manuscript: 

“The target sample was calculated using the G * Power 3.1.9.7. With α = 0.05, power = 0.95, and an effect size = 0.2, the required sample size was 262. A total of 450 women we invited to participate in the study from which 299 agreed to participate. The response rate was 66.5%.”

  1. You report “significant” differences, which appears statistically true, but the differences are not that large. Can you discuss this a bit more? This study may say almost as much about non-refugee women as it does about refugees. Response: This is an excellent point raised by the reviewer. This is now added in the revised manuscript.
  2. Tables 2 and 3. Can this be depicted in graphic form? It will be difficult for most readers to sift through the statistics and make sense of it. A visual snapshot of what these numbers mean would be helpful.

Response: While we agree that a graphical representation would simplify data comprehension for the readers, but unfortunately linear regression and correlation analyses cannot be accurately and purposefully depicted and/or illustrated in a graphical form, hence routinely presented in a tabular form.

Discussion

  1. Overall good and focussed enough to capture the essential points. Good comments on the importance of health care insurance. Response: Thank you.
  2. The sentence: “When opportunities of higher education…value of education” is bit confusing. I suggest reworking this sentence structure. Response: The sentence has been revised as suggested.
  3. I am glad you highlight that both the refugee and host women score high on the SRQ. You highlight a number of women’s issues but do not comment on sex and gender-based violence. Although not the focus of your study, you may wish to acknowledge it in your introduction and/or discussion.                                                       Response: Done as recommended.
  4. One of the limitations of your study is that it cannot capture all the relevant factors affecting women’s mental health.                                                                                           Response: We agree with reviewer’s comment. We have now added this as a limitation of our study.
  5. References seem to draw on culturally relevant literature  

Response: Thank you.

Reviewer 2 Report

Thank you for allowing me to review this very interesting study that investigated and compared the mental health and general well-being of Syrian refugee women resettled in Jordan and Jordanian women. The paper was overall well-written, however, the methodology could be improved for transparency. Below are my detailed comments.

Introduction

·       The background is well-written. However, the paper would benefit from adding more about the context. For example, what are the policies for refugees settling in Jordan? Do they get a residence permit? Are they living in refugee camps? You mentioned studies conducted with Syrian refugee women resettling in different settings. Are there any studies which investigated Syrian refugees in Jordan?

·       Since you are comparing with Jordanian women, are there studies that investigated Jordanian women’s mental health?

·       The last paragraph of the introduction, page 3, line 58-66, can be moved to the methods. However, you need to argue why general health is included in the study, i.e., the relationship between general health and mental health.

Methods

·       Please describe the setting. In the urban city you conducted in your study, how many refugees from Syria are living? Are they living in refugee camps?

·       Please describe how the participants were recruited. For example, you mentioned this network; what is this local Jordan community network? Explain more and how the Syrian refugee women were recruited and where they were recruited from?

·       How have you used snowballing?

·       How did you end up with this sample size? Have you conducted any power calculations?

Results

·       The result is well-written

Discussion, conclusion and limitation

·       Both Jordanian and Syrian refugee women scored higher than the clinical cut-off of SRQ. However, in your discussion and conclusion, you have not highlighted the implication for practice for Jordanian women. Is there a prevention program that can be put in Jordanian primary healthcare?

·       Please describe the limitation regarding the measurements used in the study.

·       Elaborate more on the limitation of sample size.

All the best

Author Response

REVIEWER#2

Thank you for allowing me to review this very interesting study that investigated and compared the mental health and general well-being of Syrian refugee women resettled in Jordan and Jordanian women. The paper was overall well-written, however, the methodology could be improved for transparency. Below are my detailed comments.                                                                                                                                                        Response: Thank you.

Introduction

  1. The background is well-written. However, the paper would benefit from adding more about the context. For example, what are the policies for refugees settling in Jordan? Do they get a residence permit? Are they living in refugee camps? You mentioned studies conducted with Syrian refugee women resettling in different settings. Are there any studies which investigated Syrian refugees in Jordan?

Response: The Introduction was revised to include points highlighted in the comments.

  1. Since you are comparing with Jordanian women, are there studies that investigated Jordanian women’s mental health?

Response: Some studies that investigated Jordanian women’s mental health were added to the Introduction section.

  1. The last paragraph of the introduction, page 3, line 58-66, can be moved to the methods. However, you need to argue why general health is included in the study, i.e., the relationship between general health and mental health.

Response: The mentioned paragraph was revised to include the recognized association between mental health and chronic diseases in the literature.

Methods

  1. Please describe the setting. In the urban city you conducted in your study, how many refugees from Syria are living? Are they living in refugee camps?

 Response: The required information about the setting was added to the Method section.

  1. Please describe how the participants were recruited. For example, you mentioned this network; what is this local Jordan community network? Explain more and how the Syrian refugee women were recruited and where they were recruited from?                    

    Response: Done as recommended

  1. How have you used snowballing? How did you end up with this sample size? Have you conducted any power calculations?         

Response: The required information was added to the Method section.

Results

  1. The result is well-written

Response: Thank you.

Discussion, conclusion and limitation

  1. Both Jordanian and Syrian refugee women scored higher than the clinical cut-off of SRQ. However, in your discussion and conclusion, you have not highlighted the implication for practice for Jordanian women. Is there a prevention program that can be put in Jordanian primary healthcare?

Response: We have discussed this result in the last paragraph of the discussion, we have now elaborated this point in the revised manuscript.

  1. Please describe the limitation regarding the measurements used in the study.

Response: Done as recommended.

  1. Elaborate more on the limitation of sample size.

Response: Sample size calculation was added to the Method section. The point regarding sample size was deleted from the limitation paragraph.

Author Response

REVIEWER#3

Thank you for giving me the opportunity to review this manuscript which compared mental health status of Syrian refugee women with local Jordanian women. Despite the interesting research gap which this study attempted to fill, there were several comments which the authors will need to work on before this manuscript can be reconsidered for publication.

  1. Please be consistent. Some "p"s were italicized (check the results section as well). please check the journal guidelines on how to write p value.

Response: Done as recommended.

  1. The last sentence of the abstract should be a conclusion of whether the findings of the study achieve the study objectives and what should be recommended based on the findings.

Response: The conclusion was revised as suggested.

  1. How many Syrian women among these refugees?

Response: An estimate of the number was added to the text.  

  1. This is irrelevant statement. I would suggest to eliminate it.

Response: Done as recommended.

  1. Authors are recommended to elaborate more on this statement. explain what are the social stressors usually experienced by refugees.

Response: The statement has been elaborated as recommended.

  1. Please add where this review was conducted

Response: Done as recommended.

  1. Authors are recommended to add a comment here about previous reports in the literature addressing these kinds of challenges among Syrian refugees in Jordan or other refugees such as Palestinian, Iraqi...etc

Response: This is an excellent suggestion. Done as recommended.

  1. Authors are recommended to mention these details in the methodology section.

Response: Done as recommended.

  1. I found the following methodological aspects deficient: 1-Sample size estimation and power analysis.

Response: More details on the sample size estimation and power analysis were added to the Method section as suggested.

  1. How many women were invited to participate in the study? How many were excluded? What were the reason(s) for exclusion?

Response: The required information was added to the Method section.

  1. Were there any missing data? How missing data were treated- what type of imputation method was used? What was the response rate? All these data/information were not described.

Response: All the statistical analyses were conducted using complete case analyses, and the response rates of variables of interest ranged from 96% to 100%.  

  1. Timing of the study and it is relation to COVID-19 pandemic. if the study was conducted during or after COVID-19 pandemic, authors are recommended to justify that the mental health status of refugees has been affected by post-migration status only and COVID-19 pandemic effect has been controlled.

Response: As mentioned in the first paragraph of the results section, this study was conducted between July and September 2022.  Therefore, the study was conducted after the peak COVID-19 pandemic period. Also, our targeted population, the Syrian refugee women and the local Jordanian families, both resided in the same urban areas with the pandemic impacting both groups similarly.

  1. The authors have not described how the questions were phrased for assessment of demographic information and mental health status (for exam[le yes/no questions, multiple choice questions, Likert-scale question types...etc)

Response: Mental health status was evaluated via Afghan Symptom Checklist (ASC) and Self reporting Questionnaire (SRQ).

  1. My understanding is that this study was conducted among Syrian refugees in A Za'atari camp. Is this correct? If yes, please mention this clearly throughout the manuscript. Also, there are another camp in Jordan which hosts a large number of Syrian refugees called Azraq camp, authors are recommended to justify why only refugees in Ar-Ramtha area has been recruited in this study.

Response: The study was conducted on urban Syrian refugee women living within the local community and not in refugee camps. This was explained in the revised manuscript.

  1. Authors are recommended to list the references of researches that have validated the Arabic version of the questionnaires listed.

Response: The Arabic versions of the surveys were validated in previous studies as well as in our previously published papers.

The following references were cited in the manuscript:

  1. Atrooz, F., et al., Displacement and Isolation: Insights from a Mental Stress Survey of Syrian Refugees in Houston, Texas, USA. International Journal of Environmental Research and Public Health, 2022. 19(5): p. 2547.
  2. Almadi, T., et al., An Arabic version of the perceived stress scale: translation and validation study. Int J Nurs Stud, 2012. 49(1): p. 84-9.
  3. El-Rufaie, O.E. and G.H. Absood, Validity study of the Self-Reporting Questionnaire (SRQ-20) in primary health care in the United Arab Emirates. International Journal of Methods in Psychiatric Research, 1994.
  4. Al-Subaie, A.S., K. Mohammed, and T. Al-Malik, The Arabic self-reporting questionnaire (SRQ) as a psychiatric screening instrument in medical patients. Annals of Saudi medicine, 1998. 18(4): p. 308-310.

  1. The findings of the study described in the text were duplication of what were already mentioned in the tables. In fact, the authors should summarize the findings and include only the salient features of the study rather than repeating what already presented in the tables.

Response: The result section was revised as suggested.

  1. Please see my comment above about the timing of the study and impact of COVID-19 pandemic

Response: As mentioned in the first paragraph of the results, this study was conducted between July and September 2022.

  1. Be consistent (round/square brackets). please check the hole table. I found several of this typo.

Response: In the table, we used square brackets for [%] for N number, while round brackets were used for (SD) of the mean and as depicted in the columns title.

  1. Elaborate more about the pre-migration traumatic stress and support this statement with a reference.1-support this statement with a reference.

Response: Done as recommended.

  1. What is the percentage of Syrian refugee females attending schools in Jordan ?

Response: It is estimated that Syrian refugee females attending schools in Jordan is less than 60%. 

  1. Discuss the quality and affordability of mental health services in Jordan for both Jordanian and non-Jordanian residents.

Response: This is an excellent suggestion. We have now incorporated this in the revised manuscript. There are few mental health institutions in Jordan, there are only three mental health hospitals for adults and one specialized psychiatric hospital for children, with total number of psychiatrists in Jordan not exceeding 2 per 100 000 residents. This point was included in the discussion section in the revised manuscript.